# Elevated CSF LRG and Decreased Alzheimer’s Disease Biomarkers in Idiopathic Normal Pressure Hydrocephalus

**DOI:** 10.3390/jcm10051105

**Published:** 2021-03-06

**Authors:** Aleksi Vanninen, Madoka Nakajima, Masakazu Miyajima, Tuomas Rauramaa, Merja Kokki, Tadeusz Musialowicz, Petra M. Mäkinen, Sanna-Kaisa Herukka, Anne M. Koivisto, Juha E. Jääskeläinen, Mikko Hiltunen, Ville Leinonen

**Affiliations:** 1Department of Neurosurgery, Kuopio University Hospital, FI-70029 Kuopio, Finland; aleksi.vanninen@uef.fi (A.V.); juha.e.jaaskelainen@kuh.fi (J.E.J.); 2Neurosurgery, Institute of Clinical Medicine, University of Eastern Finland, FI-70210 Kuopio, Finland; 3Department of Neurosurgery, Juntendo University School of Medicine, Tokyo 113-8342, Japan; madoka66@juntendo.ac.jp; 4Department of Neurosurgery, Juntendo Tokyo Koto Geriatric Medical Center, Tokyo 136-0075, Japan; mmasaka@juntendo.ac.jp; 5Department of Pathology, Kuopio University Hospital, FI-70029 Kuopio, Finland; tuomas.rauramaa@kuh.fi; 6Pathology, Institute of Clinical Medicine, University of Eastern Finland, FI-70210 Kuopio, Finland; 7Department of Anaesthesia and Intensive Care Medicine, Kuopio University Hospital, FI-70029 Kuopio, Finland; merja.kokki@kuh.fi (M.K.); tadeusz.musialowicz@kuh.fi (T.M.); 8School of Medicine, University of Eastern Finland, FI-70210 Kuopio, Finland; 9Institute of Biomedicine, University of Eastern Finland, FI-70210 Kuopio, Finland; petra.makinen@uef.fi (P.M.M.); mikko.hiltunen@uef.fi (M.H.); 10Department of Neurology, Kuopio University Hospital, FI-70029 Kuopio, Finland; sanna-kaisa.herukka@uef.fi (S.-K.H.); anne.koivisto@kuh.fi (A.M.K.); 11Neurology, Institute of Clinical Medicine, University of Eastern Finland, FI-70210 Kuopio, Finland; 12Department of Neurosciences, University of Helsinki, FI-00014 Helsinki, Finland; 13Department of Geriatrics, Helsinki University Hospital, FI-00014 Helsinki, Finland

**Keywords:** hydrocephalus, normal pressure, cerebrospinal fluid, Alzheimer’s disease, biopsy, brain

## Abstract

Leucine-rich-alpha-2-glykoprotein (LRG) is suggested as a potential biomarker for idiopathic normal pressure hydrocephalus (iNPH). Our goal was to compare the cerebrospinal fluid (CSF) LRG levels between 119 iNPH patients and 33 age-matched controls and with the shunt responses and the brain biopsy Alzheimer’s disease (AD) pathology among the iNPH patients. CSF LRG, Aβ_1-42_, P-tau_181_, and T-tau were measured by using commercial ELISAs. The LRG levels in the CSF were significantly increased in the iNPH patients (*p* < 0.001) as compared to the controls, regardless of the AD pathology. However, CSF LRG did not correlate with the shunt response in contrast to the previous findings. The CSF AD biomarkers, i.e., Aβ_1-42_, T-tau, and P-tau correlated with the brain biopsy AD pathology as expected but were systematically lower in the iNPH patients when compared to the controls (<0.001). Our findings support that the LRG levels in the CSF are potentially useful for the diagnostics of iNPH, independent of the brain AD pathology, but contrary to previous findings, not for predicting the shunt response. Our findings also suggest a need for specific reference values of the CSF AD biomarkers for the diagnostics of comorbid AD pathology in the iNPH patients.

## 1. Introduction

Normal pressure hydrocephalus (NPH) is a neurological disease with a classic triad of symptoms including (1) impaired cognition, (2) gait difficulty, and (3) urinary incontinence [1]. NPH is idiopathic (iNPH) when no known predisposing factors, such as trauma or hemorrhage, are present [2]. The surgical insertion of a shunt to bypass the normal cerebrospinal fluid (CSF) flow route eases the symptoms for most patients [3], but, unfortunately, the improvement of the symptoms varies considerably. Furthermore, long-term improvements of the shunt surgery may decline over the years [4]. A significant number of initially shunt-responsive patients develop either clinical dementia or mild cognitive impairment in relation to the extent of cognitive impairment prior to the shunt surgery [5]. Despite the symptoms of iNPH being alleviated by the shunt surgery, the etiology and the pathogenesis of the disease remain unknown. The most well-documented risk factors for iNPH are hypertension and Type II diabetes mellitus [6]. There are multiple theories regarding the pathogenesis of iNPH, the abnormality of the CSF dynamics, the reduced metabolism, the vascular irregularities, and the inflammation [7].

Differential diagnostics between iNPH and Alzheimer’s disease (AD) are important, as the clinical symptoms may overlap. AD is the most common form of clinical dementia within the elderly population [8] and is also often comorbid in the iNPH patients (iNPH-AD) [9] as predicted by the brain biopsies taken during shunt surgeries [10]. The presence of AD pathology has been shown to significantly reduce the long-term success of the shunt in iNPH even though short-term improvements can be seen in the patients with AD-associated CSF profiles [11].

Leucine rich alpha-2 glycoprotein (LRG) is a protein first isolated from human serum by Haupt and Baudner [12]. The LRG expression is upregulated by an inflammatory response and the concentration of LRG increases with aging and the decline of cognitive function [13,14]. It has also been shown that the levels of LRG in the CSF of the iNPH patients are significantly elevated when compared to a control group consisting of people with headaches and no other complications from neurological conditions [15]. The CSF levels of LRG have been shown, together with the tau protein levels, to predict the shunt responses in the iNPH patients. Briefly, higher levels of CSF LRG, together with a positive tap test result, predicted a successful shunt response, while lower levels of LRG, with higher levels of tau, predicted a negative shunt response [16].

This study aims to decipher the diagnostic value of CSF LRG by comparing cognitively healthy controls to the iNPH patients. Furthermore, within the iNPH group, the goal was to evaluate the effects of CSF LRG on the shunt responses and the AD pathology from the brain biopsies. Our hypothesis was that elevated CSF LRG indicates iNPH but does not correlate with the AD pathology seen in the brain biopsies.

## 2. Materials and Methods

### 2.1. Patient Recruitment

Patients presenting 1 to 3 of the core symptoms possibly relating to iNPH (gait difficulty, impaired cognition, or urinary incontinence), together with an MR image featuring a disproportionate ratio of the brain ventricles (Evans index > 0.3) to the sulci of the cerebral convexities [2], were evaluated in the Department of Neurosurgery, Kuopio University Hospital (KUH), which provides acute and elective neurosurgical services to a defined catchment population of 815,000 people in Eastern Finland. Our patient cohort included 119 consecutive patients who were shunted due to probable iNPH [17], diagnosed according to a previously published protocol [18] between 2012 and 2019. The outcome of the shunt surgery was evaluated at the three-month follow-up, clinically and by iNPH grading scale (iNPHGS) [5,19]. All patients with suspected iNPH from the KUH catchment population are included in the Kuopio NPH Registry featuring the clinical follow-up outcome data and other hospital diagnoses, medications, and the causes of death with the autopsy reports (www.uef.fi/nph (accessed on 4 March 2021)). CSF samples were taken prior to shunt.

### 2.2. Frontal Cortical Biopsy Sampling

The biopsy procedure was performed as described previously [18]. A 12-mm burr hole was made to the right frontal cortexes, approximately 3 cm laterally from the midline. One to three cylindrical cortical biopsy samples of 2 to 5 mm in diameter and 3 to 10 mm in length were then obtained with a biopsy needle prior to the insertion of the intraventricular catheter for the CSF shunt. Part of the sample was placed in buffered formalin and then embedded in paraffin after an overnight fixation. The biopsy samples were then sliced into 7-µm thick sections, which were afterwards stained with hematoxylin-eosin (HE) and immunohistochemistry (IHC), including AT8, p62, and Aβ [6F/3D, labelling both parenchymal aggregates as well as cerebral amyloid angiopathy (CAA), i.e., the cored plaques, the vascular amyloid, and the stellate and diffuse plaques seen at the early stages of the disease]. The stained sections were assessed under a light microscope at 100–200× magnification. The Pτ structures, either cellular or neuritic, were identified and then rated as either negative or positive. The stellate, diffused, and compact plaques were assessed in the sections stained with Aβ-IHC and the staining results of Aβ were semi-quantitatively rated (Dr. Rauramaa) [18,20,21]. Brain biopsy samples were obtained from all patients included in the study. Patients were divided into groups according to AD-pathology shown in immunohistochemistry analysis as No-AD (No AD-pathology), Aβ (Amyloid pathology), or Aβ + Tau (Both amyloid- and tau-pathologies). 

### 2.3. The CSF Sampling

Lumbar CSF samples were harvested from the iNPH patients during the diagnostic tap test. The CSF samples from the control group were gathered from patients attending a knee replacement surgery with spinal anesthesia in KUH [22]. The CSF samples were centrifuged and then stored in polypropylene tubes at −80 °C until the analyses were conducted in the UEF Biomarker Laboratory and the Department of Biomedicine according to the standardized protocols. The CSF levels of LRG, A-β_1-42_, T-tau, and P-tau_181_ were measured by using commercial ELISA (Enzyme-linked-immunosorbent-assay) kits (Human LRG Assay, IBL, Japan, Innotest β-amyloid1–42, Innotest Tau-Ag, Innotest Phosphotau (181P), Fujirebio, Ghent, Belgium), using the manufacturer’s protocol. The cut-off values, considered normal, for the CSF samples were as follows: Aβ_1-42_ > 500 mg/mL, T-tau < 400 pg/mL, and P-tau_181_ < 70 pg/mL.

### 2.4. Statistical Analysis

The data was analyzed by using the SPSS software (Version 22. Inc., Chicago, IL, USA). The univariate analysis of covariance (ANCOVA), using sex and age as the covariates, was used to analyze the differences between the groups for each individual CSF marker level. Out of the investigated biomarkers, only LRG was not normally distributed but, since the number of cases in the group was over 30 (except iNPH Aβ + Tau group), we considered that parametric tests could be used. Age and Mini Mental State Examination (MMSE) between the groups were investigated with *t*-tests on independent samples. For pairwise comparisons between the groups with ANCOVA, a sidak method of multiple comparisons was used. Binary logistic regression was used to create a multivariate model between the control and the iNPH groups. The final model was picked by removing the variables with the highest *p*-values until all variables were significant. The receiver operating characteristic (ROC) curve was used to test the function of the model. Multinomial logistic regression was used to create a model for the different AD pathologies. The initial model featured post-operative (3 months) MMSE scores, NPHGS-baseline scores, and three-month follow-up NPHGS scores as well as the statistically significant biomarkers from the univariate tests. Data was available for 108 out of 119 patients. The initial model was then modified by removing the variables with the highest *p*-values. LRG was kept in the analyses even when not significant, as it was the main interest of this study. ANCOVA was also used for the analysis of the relation of the biomarker levels to the shunt response. Multiple linear regression was used to analyze the effects of the CSF biomarker levels on the three-month change in the individual NPHGS components (gait, cognition, and incontinence).

### 2.5. Ethics Statement

The research has the permission of the Ethics Committee, Hospital District of Northern Savo, Kuopio, Finland. All subjects gave their written informed consent, and the study was conducted in accordance with the latest revision of the Declaration of Helsinki.

## 3. Results

The CSF LRG levels were increased in the iNPH patients when compared to the control group regardless of the AD pathology, as presented in Figure 1 and Table 1. This difference was statistically significant for No AD-groups and Aβ-groups, but not for the Aβ + Tau-group. The CSF AD biomarkers, i.e., Aβ_1-42_, T-tau, and P-tau_181_, correlated with the brain biopsy AD pathology as expected, but were systematically lower in the iNPH patients when compared to the controls (Figure 1). The binominal logistic model, presented in Table 2, yielded a regression equation of 0.00317 × LRG − 0.00577 × Aβ_1-42_ − 0.0518 × P-tau simplified to 1 × LRG − 2 × Aβ_1-42_ – 16 × P-tau. The ROC curve (Figure 2) for the binary model identifying the iNPH patients from the control population yielded an AUC of 0.896. Results indicate that the patients with increased LRG together with decreased Aβ_1-42_ and P-Tau_181_ are likely to have iNPH. Comparing ROC-curves for models with and without LRG shows that LRG adds a diagnostic value to the model.

LRG Leucine-rich-glycoprotein, Aβ-42 amyloid-β_1-42_. TAU total tau, P-Tau phosphorylated tau_181_, AD Alzheimer’s disease. Middle lines indicate median. Top and bottom of the boxes indicate third and first quartile, respectively. Whiskers indicate minimum and maximum values within 1.5 times the Intra-quartile value. Circles indicate individual values. The bold horizontal line indicates cut-off values for diagnostics of AD (Pathological values below the line for Aβ_1-42_ and above the line for both tau-proteins). Groups are as follows: Control = Control group of healthy individuals, No AD = iNPH patients with no AD pathology in brain biopsy, Aβ = iNPH patients with Aβ-pathology in brain biopsy, and Aβ + Tau = iNPH patients with both amyloid and tau pathology in brain biopsy.

AUC Area Under Curve. AUC_1_ = AUC for model featuring LRG, Aβ_1-42_, and P-Tau_181_. AUC_2_ = AUC for model featuring Aβ_1-42_ and P-Tau_181_.

The analyses between the iNPH patients in relation to the brain biopsy AD pathology are presented in Table 3. The final multinomial logistic regression model (Table 4) yielded an expected association between the brain biopsy AD pathology and CSF Aβ_1-42_ and P-tau, but no significant effects on CSF LRG.

None of the investigated CSF biomarkers yielded statistical significance for predicting short-term shunt responses as presented in Table 5. However, brain biopsy Aβ and Tau together indicated a worse shunt outcome compared with Aβ only or no AD-related pathology.

## 4. Discussion

We found out that CSF LRG is a potential biomarker of probable iNPH [17] and combining the LRG levels with the Aβ_1-42_ and P-tau levels is a very sensitive model (95.8%) in separating the cognitively healthy subjects from the iNPH patients, but lacks specificity (54.5%). This lack of specificity is most likely due to the nature of LRG being a pro-inflammatory protein and because the levels of LRG also increase in the CSF due to a normal aging process. When comparing the AD pathologies of the iNPH patients in our study population, we found that the levels of LRG are equal between the studied groups while the levels of Aβ_1-42_, T-tau, and P-tau_181_ are useful in separating the iNPH patients with AD-related pathology from those with no AD-related pathology. The patients exhibiting tau pathology alone were left out of the analyses between the different AD-related pathologies due to the small sample size (*n* = 3), but further investigation into the patients with tau pathology would be of interest due to the significant differences in the individual CSF profiles when compared to the rest of the study population. This suggests that the patients with tau-only pathology were afflicted by other neurological diseases than AD.

It is interesting that, in our study population, none of the investigated CSF biomarkers correlated with the short-term shunt responses, hinting that the investigated biomarkers are not credible in predicting immediate shunt responses. The current result contrasts the findings of Nakajima et al. where the CSF LRG levels correlated with the shunt response in the 12-month follow-up [16]. However, the brain AD-pathology in the brain biopsy indicated a poor shunt outcome, as seen previously in a one-year follow-up [23] suggesting that evaluation of the potential prognostic value of CSF LRG in a further study with longer follow-up would be of interest.

The reduced levels of the CSF AD biomarkers are already associated with iNPH. Reduced total tau and phospho-tau may indicate reduced cortical metabolism, reduced neuronal activity, decreased clearance, or simply be due to dilution caused by increased CSF volume in iNPH [24,25,26]. Our results support the previous findings and also confirm the association between the brain AD pathology and iNPH [20]. These, together, can motivate to define the specific reference values for the iNPH patients in detecting comorbid AD. Elevated LRG levels may reveal a more specific iNPH-related pathological process, suggesting that CSF LRG could be used as a biomarker in the diagnosis of iNPH. The relation between the CSF LRG levels and the brain AD pathology have not been previously compared. Our study clearly indicated that the LRG levels are not affected by the patients’ AD pathology.

The strengths of our study were the notable sample size of iNPH patients whose data and samples were gathered with identical protocol together with the cognitively tested healthy controls. All the CSF samples were analyzed at the same time. Unfortunately, the limited number of tau-positive, amyloid-negative patients (*n* = 3) prevented us from including them in the analysis between the AD groups. These patients most likely have other neurological disorders than AD.

Future research should be focused on finding minimally invasive means to predict the shunt responses in iNPH. LRG requires further research as our results partly contradicted the former results in predicting the shunt responses, which could be explained by short follow-up and the small sample size of the shunt non-responders. Despite this, our shunt non-responder sample size is one of the largest investigated to date. In the future, combining LRG with other promising biomarkers, such as protein tyrosine phosphatase receptor type Q (PTPRQ) [27,28], could help determine the shunt response with a single CSF sample, but, as of now, more laborious procedures are still required to accurately predict the shunt response. 

Overall, our study suggests that CSF LRG has potential as a diagnostic biomarker of iNPH, but still requires further validation. Furthermore, iNPH patients need independent reference values of the established CSF AD biomarkers [29] for the diagnosis of comorbid AD due to the unclear mechanism causing the decreased CSF biomarker levels among the iNPH patients.

## 5. Conclusions

LRG is a sensitive biomarker for probable iNPH but cannot be used alone to predict a shunt response. The diagnostic value of the CSF AD biomarkers could be augmented by specific reference values for the INPH patients.

## Figures and Tables

**Figure 1 jcm-10-01105-f001:**
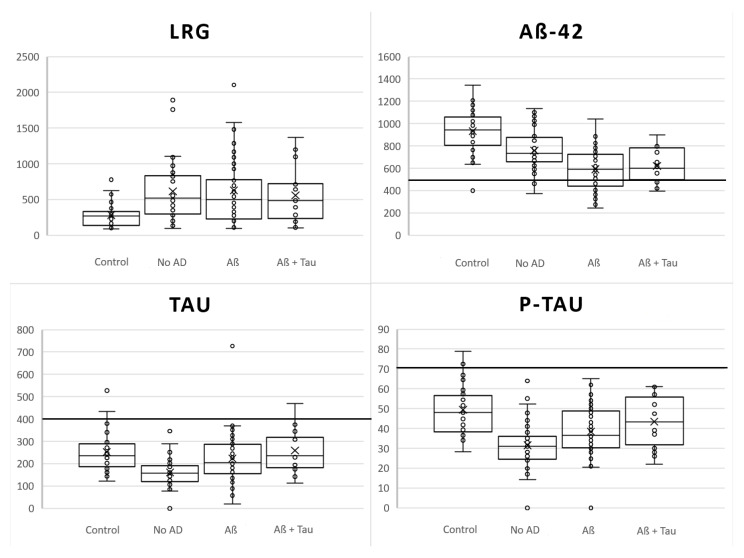
Boxplots for investigated cerebrospinal fluid (CSF) biomarker levels between study populations.

**Figure 2 jcm-10-01105-f002:**
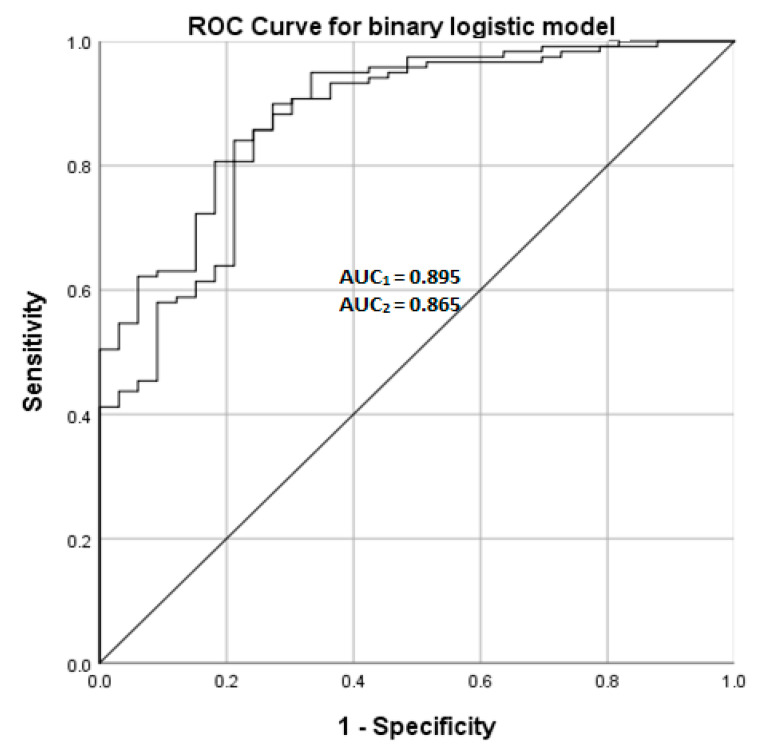
Receiver operating characteristic (ROC) curve for a binary logistic model using LRG, Aβ_1-42_, and P-Tau_181_ (AUC_1_) and model with Aβ_1-42_ and P-Tau_181_ (AUC_2_).

**Table 1 jcm-10-01105-t001:** Patient demographics.

Category		Control	iNPH No-AD	iNPH Aβ	iNPH Aβ+Tau	Pooled iNPH
*n*		33	48	52	16	119
Age	Mean (SD)	73 (4)	73 (6)	76 (5) *	81 (3) **	75 (6) *
MMSE	Median (IQR)	28 (27–29)	24 (22–27) **	23 (18–26) **	20 (16–24) **	24 (19–26) **
LRG pg/mL	Mean (SD)	283.0 (170.5)	614.1 (418.1) *	625.3 (493.0) *	558.6 (384.3)	610.1 (440.7) **
Aβ1-42 pg/mL	Mean (SD)	933.2 (194.9)	757.3 (167.7) **	591.5 (182.3) **	622.0 (153.8) **	658.8 (189.4) **
Tau pg/mL	Mean (SD)	252.3 (89.2)	160.3 (59.5) **	223.7 (110.3)	259.1 (95.2)	199.7 (97.3) **
P-Tau181 pg/mL	Mean (SD)	49.4 (12.5)	31.5 (11.4) **	38.4 (14.1) **	43.4 (13.1)	35.5 (14.2) **

iNPH Idiopathic Normal Pressure Hydrocephalus, SD standard deviation, AD Alzheimer’s disease, Aβ_1-42_ Amyloid-β_1-42_, P-Tau_181_ phosphorylated tau-protein, MMSE mini mental state exam, and LRG leucine rich glycoprotein. For three patients, pre-operative MMSE scores were missing and were filled with three-month follow-up scores. For one patient, CSF LRG was unavailable and replaced with mean LRG levels of 118 patients * *p*-value for biomarkers from ANCOVA controlled for age and sex. *p*-values for age and MMSE from independent samples *t*-test. * indicates 0.05 > *p* > 0.001 ** indicates *p* < 0.001. Three patients with Tau only pathology were excluded from the table due to low *n*. Groups are as follows: Control = Control group of healthy individuals, Pooled iNPH = combined group of all iNPH patients, iNPH No AD = iNPH patients with no AD pathology in brain biopsy, iNPH Aβ = iNPH patients with Aβ-pathology in brain biopsy, and iNPH Aβ + Tau = iNPH patients with both amyloid and tau pathology in brain biopsy.

**Table 2 jcm-10-01105-t002:** Binary logistic regression results for a model with and without LRG.

Variable	Regression Coefficient	Odds Ratio	95% C.I. for OR
LRG *	0.00317	1.00318	1.000682–1.00567
Aβ1-42 *	−0.00577	0.994	0.991–0.997
P-Tau181 *	−0.0518	0.950	0.910–0.990
Excluded variables **			
Age	0.091	1.095	0.978–1.227
Model value1	**Specificity**	**Sensitivity**	**ROC AUC (95% C.I.)**
	54.5%	95.8%	0.895 (0.837–0.952)
Variable	**Regression Coefficient**	**Odds Ratio**	**95% C.I for OR**
Aβ1-42 *	−0.00632	0.994	0.991–0.996
P-Tau181 *	−0.0542	0.947	0.910–0.986
Excluded variables **			
-			
Model value2	**Specificity**	**Sensitivity**	**ROC AUC (95% C.I.)**
	48.5%	95.8%	0.865 (0.795—0.934)

* *p*-value < 0.05. ** Exclusion criterion *p*-value > 0.05. C.I. confidence interval. OR odds ratio. LRG Leucine rich glycoprotein. Aβ_1-42_ Amyloid-β_1-42_. P-Tau_181_ Phosphorylated tau-protein. AUC area-under-curve.

**Table 3 jcm-10-01105-t003:** Cerebrospinal fluid (CSF) biomarker levels according to brain biopsy Alzheimer’s disease (AD)-related pathology in iNPH patients.

Category		No-AD	Aβ	*p* *	Aβ and Tau	*p* *	Tau **
*n*		48	52		16		3
Age	Mean (SD)	73 (6)	76 (5)		81 (3)		71 (2)
MMSE	Median (IQR)	24 (22–27)	23 (18–26)		20 (16–24)		20 (14–22)
LRG pg/mL	Mean (SD)	614.1 (418.1)	625.3 (493.0)	0.98	558.6 (384.3)	0.71	557.2 (120.4)
Aβ1-42 pg/mL	Mean (SD)	757.3 (167.7)	591.5 (182.3)	<0.001	622.0 (153.8)	0.059	445.8 (44.7)
Tau pg/mL	Mean (SD)	160.3 (59.5)	223.7 (110.3)	0.015	259.1 (95.2)	0.71	96.1 (26.7)
P-Tau181 pg/mL	Mean (SD)	31.5 (11.4)	38.4 (14.1)	0.067	43.4 (13.1)	0.032	6.3 (11.0)
NPHGS-baseline	Median (IQR)	6 (4–8)	7 (4–10)		8 (4–9.5)		10 (10–12)
NPHGS-change	Median (IQR)	−1 (−3–0)	−1 (−2–0)		1 (−1–2)		−1 (-2–1)

iNPH Idiopathic Normal Pressure Hydrocephalus, SD standard deviation, AD Alzheimer’s disease, Aβ_1-42_ Amyloid-β_1-42_, P-tau_181_ phosphorylated tau-protein, MMSE mini-mental state exam, LRG leucine-rich glycoprotein, NPHGS Normal pressure hydrocephalus grading scale, NPHGS baseline preoperative NPHGS score for iNPH patients, NPHGS change difference between preoperative NPHGS, and 3-month postoperative NPHGS score. * p-values are obtained from Analysis of covariance ANCOVA controlled for age, Mini Mental State Examination (MMSE), and sex. ** Patients in this category were not used for statistical analysis due to low n. Groups are as follows: No AD = iNPH patients with no AD pathology in brain biopsy, Aβ = iNPH patients with Aβ-pathology in brain biopsy, Aβ + Tau = iNPH patients with both amyloid and tau pathology in brain biopsy, and Tau = iNPH patients with only Tau-pathology.

**Table 4 jcm-10-01105-t004:** Multi-nominal logistic regression results for Alzheimer’s disease (AD)-pathologies.

Category		Likelihood Ratio Test		Parameter Estimates	
				Vs Aβ-Pathology	Vs Aβ + Tau
Parameter	−2 Log	Chi-Square	*p*	*p*	*p*
Variable					
LRG	156.6	1.1	0.58	0.325	0.876
Aβ1-42	193.3	37.7	<0.001	<0.001	<0.001
P-tau181	180.3	24.8	<0.001	<0.001	<0.001
iNPHGS-change	164.9	9.4	0.009	0.90	0.009

LRG leucine-rich-glycoprotein, Aβ Amyloid-β_1-42_. P-tau_181_ phosphorylated Tau. NPHGS-Change Change between preoperative and three-month postoperative NPHGS scores.

**Table 5 jcm-10-01105-t005:** CSF biomarker levels according to the shunt response.

Category		Non-Responder	Responder	*p* *
*n*		27	91	
Age	Mean (SD)	77 (4)	75 (6)	0.031
MMSE	Median (IQR)	21 (16–26)	24 (20–27)	0.051
LRG pg/ml	Mean (SD)	652.7 (439.0)	599.0 (445.1)	0.636
Aβ1-42 pg/ml	Mean (SD)	617.5 (172.7)	673.9 (192.2)	0.263
Tau pg/ml	Mean (SD)	211.3 (75.9)	190.4 (86.8)	0.822
P-tau181 pg/ml	Mean (SD)	38.3 (12.2)	34.7 (14.8)	0.683
NPHGS-baseline	Median (IQR)	8 (4–10)	6 (4–9)	
NPHGS-change	Median (IQR)	0 (−1–1)	−1 (−3–0)	

iNPH Idiopathic Normal Pressure Hydrocephalus, SD standard deviation, AD Alzheimer’s disease, A-β Amyloid-β_1-42_, P-Tau_181_ phosphorylated tau-protein, MMSE mini mental state exam, LRG leucine-rich glycoprotein, NPHGS Normal pressure hydrocephalus grading scale, NPHGS baseline preoperative NPHGS score for iNPH patients, NPHGS change difference between preoperative NPHGS, and 3-month postoperative NPHGS score. * *p*-value is obtained from ANCOVA controlled for age and sex.

## Data Availability

Raw data cannot be made publicly available due to patient confidentiality but can be provided in research use upon reasonable request and appropriate academic collaboration agreement notifying data protection.

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
