# Peer review of "Elevated CSF LRG and Decreased Alzheimer’s Disease Biomarkers in Idiopathic Normal Pressure Hydrocephalus"

_jcm, 2021, doi:10.3390/jcm10051105_

Round 1
Reviewer 1 Report
This is a straightforward study of CSF biomarkers in the context of NPH. There is much similar literature, and although this study is unlikely to change clinical care, its results should be interesting to those in the field.
Author Response
We appreciate the encouraging comment.
Reviewer 2 Report
The work by Vanninen and colleagues investigates the potential diagnostic value of LRG protein in the CSF for iNPH. The main findings are that CSF levels of LRG were significantly increased in iNPH patients as compared with control individuals. The article is well-written and presented study utilizes relatively valuable samples, involving a large sample size for iNPH.
The authors use parametric tests such as ANCOVA or t-tests, but do not mention that the data was normally distributed, what is required prior conducting such types of statistical analysis.
Authors divide the iNPH patients into three groups in the figure 1, but then combine them into one group in table 1, and only here the overall significance is stated. Since the authors state that CSF LRG levels are increased in the iNPH patients when compared with controls regardless of the AD pathology, significances should be stated for each of the three groups separately when compared with the control group.
How would the authors explain the facts that CSF levels of tau and p-tau are decreased in iNPH with AD pathologies compared with healthy controls? High CSF levels of total and p-tau protein are widely known biomarkers for AD pathology.
Author Response
The work by Vanninen and colleagues investigates the potential diagnostic value of LRG protein in the CSF for iNPH. The main findings are that CSF levels of LRG were significantly increased in iNPH patients as compared with control individuals. The article is well-written and presented study utilizes relatively valuable samples, involving a large sample size for iNPH.
The authors use parametric tests such as ANCOVA or t-tests, but do not mention that the data was normally distributed, what is required prior conducting such types of statistical analysis.
Important point. Out of the investigated biomarkers only LRG was not normally distributed but since the number of cases in the group was over 30 (except iNPH Aβ + Tau group) we considered that parametric tests could be used. This is now added in statistical analysis section (lines 123-125).
Authors divide the iNPH patients into three groups in the figure 1, but then combine them into one group in table 1, and only here the overall significance is stated. Since the authors state that CSF LRG levels are increased in the iNPH patients when compared with controls regardless of the AD pathology, significances should be stated for each of the three groups separately when compared with the control group.
Table 1 is now revised and the significances included as suggested. Groups are now as follows: Control = Control group of healthy individuals, Pooled iNPH = combined group of all iNPH patients, iNPH No AD = iNPH patients with no AD pathology in brain biopsy, iNPH Aβ = iNPH patients with Aβ-pathology in brain biopsy, iNPH Aβ + Tau = iNPH patients with both amyloid and tau pathology in brain biopsy. (lines 166-168)
We also added the following sentence (lines 147-149). “This difference was statistically significant for No AD- and Aβ-groups but not for Aβ + Tau -group.” The mean values were close to equal between all the three groups of iNPH patients and lack of significance between controls and Aβ + Tau –group is considered to be due to the small sample size of the latter group.
How would the authors explain the facts that CSF levels of tau and p-tau are decreased in iNPH with AD pathologies compared with healthy controls? High CSF levels of total and p-tau protein are widely known biomarkers for AD pathology.
The following discussion is now added (lines 244-247). “Reduced t-Tau and p-Tau may indicate reduced cortical metabolism, reduced neuronal activity, decreased clearance or simply be due to dilution caused by increased CSF volume in iNPH.”
Reviewer 3 Report
The manuscript presented by Vanninen et al, explore an interesting issue, very desired goal among clinicians treating with iNPH and AD patients, which is the finding of reliable biomarkers to differentiate among both pathologies, and even predict the success that a shunt operation may have and the worth/benefit vs. complications that will have for a patient going into such surgeon process.
The work is properly written, and the Introduction section is adequate to give the background needed to introduce the problem that will be afford. However, I have several concerns regarding the Material and Methods section and more important how the results are presented.
Concerns:
- Description of the patient cohort included in the study is not clear. Were all the 119 patients included in the study diagnosed with iNPH?, under which protocol?. Were they all (119) shunted?, and then followed for 3 months and evaluated again?. If that was the case, the levels of biomarkers shown were taken previously to the shunt , or after the months follow-up protocol?.
- What does iNPHGS, stant for?. Please describe briefly.
- Sampling of brain biopsies was taken from all patients? That must be stated in the specific section.
- Results obtained from the immunohistochemistry analysis indicated in section 2.2 of Mat and Met, are not shown in any way along the manuscript and not even described in the text. That is a very precious information that is being missed, and would decorate and would make gentler and easier following the results.
- The results presented in Figure 1 and Table 1, are hard to understand, in terms of knowing exactly what patients are represented in the four different groups plotted. The different study groups must be clearly indicated in the figure legend to facilitate comprehension of the graphs. As it is now, it is not clear the meaning of No AD, AB, AB+Tau groups. This must be clearly stated in the text and legend of the figure. For instance, are they all iNPH?, whom are the controls?. Same happen in Table 3, it is not clear indicated who are the patients represented in each group.
- In Table 1, which is the correspondence with the levels presente in figure 1?. For instance, the value of 35,5 pg/ml for P-Tau shown in the table 1 for iNPH, to what group corresponds in figure 1, or there is not correspondence between data presented in figure 1 and Table 1.
- Results presented in Table 2 are poorly explained and therefore it is hard to follow the reasoning authors try to offer.
- The Conclusions must be reworded to indicate in a more straight forward the final take home message derived from the obtained results. I would expect that conclusions should somehow respond to the initial goals mentioned in the last paragraph of the introduction.
Author Response
The manuscript presented by Vanninen et al, explore an interesting issue, very desired goal among clinicians treating with iNPH and AD patients, which is the finding of reliable biomarkers to differentiate among both pathologies, and even predict the success that a shunt operation may have and the worth/benefit vs. complications that will have for a patient going into such surgeon process.
The work is properly written, and the Introduction section is adequate to give the background needed to introduce the problem that will be afford. However, I have several concerns regarding the Material and Methods section and more important how the results are presented.
Concerns:
Description of the patient cohort included in the study is not clear. Were all the 119 patients included in the study diagnosed with iNPH?, under which protocol?. Were they all (119) shunted?, and then followed for 3 months and evaluated again?. If that was the case, the levels of biomarkers shown were taken previously to the shunt , or after the months follow-up protocol?.
CSF samples were taken prior to the shunt surgery. This is now noticed in line 87. Surgical outcome was evaluated at 3 months follow-up. All iNPH patients were shunted (lines 80-81). Protocol leading to shunt surgery in described in lines 81-82.
What does iNPHGS, stant for?. Please describe briefly.
We have now added the description (lines 83-84) “iNPH grading scale (iNPHGS)”.
Sampling of brain biopsies was taken from all patients? That must be stated in the specific section.
Results obtained from the immunohistochemistry analysis indicated in section 2.2 of Mat and Met, are not shown in any way along the manuscript and not even described in the text. That is a very precious information that is being missed, and would decorate and would make gentler and easier following the results.
Brain biopsy samples were obtained from all iNPH patients included in the study. Patients were divided into groups according to the AD-pathology shown in immunohistochemistry analysis as No-AD (No AD-pathology), Aβ (Amyloid pathology) or Aβ + Tau (Both amyloid- and tau-pathologies) (lines 103-106). In addition, table 1 is revised showing now the iNPH patients grouped according to the brain biopsy pathology.
The results presented in Figure 1 and Table 1, are hard to understand, in terms of knowing exactly what patients are represented in the four different groups plotted. The different study groups must be clearly indicated in the figure legend to facilitate comprehension of the graphs. As it is now, it is not clear the meaning of No AD, AB, AB+Tau groups. This must be clearly stated in the text and legend of the figure. For instance, are they all iNPH?, whom are the controls?. Same happen in Table 3, it is not clear indicated who are the patients represented in each group.
Table 1 is now revised and the following information added. “Groups are as follows: Control = Control group of healthy individuals, No AD = iNPH patients with no AD pathology in brain biopsy, Aβ = iNPH patients with Aβ-pathology in brain biopsy, Aβ + Tau = iNPH patients with both amyloid and tau pathology in brain biopsy.” (lines 166-168)
Significances are now added: *indicates 0.05 > p > 0.001 **indicates p < 0.001. Three patients with Tau only pathology were not included in the subgroups.
Added to lines 177-182: “Groups are as follows: Control = Control group of healthy individuals, Pooled iNPH = combined group of all iNPH patients, iNPH No AD = iNPH patients with no AD pathology in brain biopsy, iNPH Aβ = iNPH patients with Aβ-pathology in brain biopsy, iNPH Aβ + Tau = iNPH patients with both amyloid and tau pathology in brain biopsy.”
Added to lines 203-206: “Groups are as follows: No AD = iNPH patients with no AD pathology in brain biopsy, Aβ = iNPH patients with Aβ-pathology in brain biopsy, Aβ + Tau = iNPH patients with both amyloid and tau pathology in brain biopsy, Tau = iNPH patients with only Tau-pathology.”
In Table 1, which is the correspondence with the levels presente in figure 1?. For instance, the value of 35,5 pg/ml for P-Tau shown in the table 1 for iNPH, to what group corresponds in figure 1, or there is not correspondence between data presented in figure 1 and Table 1.
Excellent point, we have now revised table 1 and added the following descriptions:
Lines 166-168: “Groups are as follows: Control = Control group of healthy individuals, No AD = iNPH patients with no AD pathology in brain biopsy, Aβ = iNPH patients with Aβ-pathology in brain biopsy, Aβ + Tau = iNPH patients with both amyloid and tau pathology in brain biopsy.”
Lines 177-182: “*indicates that 0.05 > p > 0.001 ** indicates p < 0.001. Three patients with Tau only pathology were excluded from the table due to low N. Groups are as follows: Control = Control group of healthy individuals, iNPH = combined group of all iNPH patients, No AD = iNPH patients with no AD pathology in brain biopsy, Aβ = iNPH patients with Aβ-pathology in brain biopsy, Aβ + Tau = iNPH patients with both amyloid and tau pathology in brain biopsy.”
Results presented in Table 2 are poorly explained and therefore it is hard to follow the reasoning authors try to offer.
We have added the following sentence to clarify this (lines 154-157): “Briefly, results indicate that patients with increased LRG together with decreased Aβ1-42 and P-Tau181 are likely to have iNPH. Comparing ROC-curves for models with and without LRG shows that LRG adds diagnostic value to the model.”
The Conclusions must be reworded to indicate in a more straight forward the final take home message derived from the obtained results. I would expect that conclusions should somehow respond to the initial goals mentioned in the last paragraph of the introduction.
We politely disagree this and have not changed the conclusions.